# Mixed Ensiling Increases Degradation Without Altering Attached Microbiota Through *In Situ* Ruminal Incubation Technique

**DOI:** 10.3390/ani15142131

**Published:** 2025-07-18

**Authors:** Xuanxuan Pu, Min Zhang, Jianjun Zhang, Xiumin Zhang, Shizhe Zhang, Bo Lin, Tianwei Wang, Zhiliang Tan, Min Wang

**Affiliations:** 1College of Animal Science and Technology, Tarim University, Alar 843300, China; puxuanxuan@taru.edu.cn; 2Key Laboratory of Tarim Animal Husbandry Science and Technology of Xinjiang Production and Construction Group, Alar 843300, China; 3Key Laboratory of Livestock and Forage Resources Utilization around Tarim, Ministry of Agriculture and Rural Affairs, Alar 843300, China; 4CAS Key Laboratory for Agro-Ecological Processes in Subtropical Region, Hunan Provincial Engineering Research Center for Healthy Livestock and Poultry Production, Institute of Subtropical Agriculture, Chinese Academy of Sciences, Changsha 410125, China; zmlalala123@outlook.com (M.Z.); zhangjianjun0522@163.com (J.Z.); zszdyxhm@126.com (S.Z.); zltan@isa.ac.cn (Z.T.); 5College of Animal Science and Technology, Guangxi University, Nanning 530004, China; linbo@gxu.edu.cn; 6State Key Laboratory of Microbial Resources, Institute of Microbiology, Chinese Academy of Sciences, Beijing 100101, China; wangtw@im.ac.cn

**Keywords:** mixed silage, rape straw, ruminal fiber degradation, micro-structure, ruminal bacteria

## Abstract

Better utilization of rape straw can provide alternative strategies for the sustainability of ruminants and food production. In the present study, rape straw was mixed ensiled with whole crop corn, and obtained good fermentation quality by enriching lactic acid bacteria. For a better understanding of mixed silage effects on the nutrition and utilization value of rape straw, rape straw was mixed with whole crop corn silage and used as a control group. Results showed that mixed silage destroyed the fiber structure of rape straw, and enhanced *in situ* nutrient degradation by increasing both rapidly and potentially degradable nutrients. In summary, mixed silage provides a potential strategy to improve nutrient and utilization value of rape straw.

## 1. Introduction

About 4.05 million tons of rape straw are produced annually in China, and its disposal has become a concern [1]. Burning, as a traditional way to treat rape straw, causes both environmental pollution and resource waste [1]. Rape straw can be used as a potential feedstuff for ruminants, but the high lignified fiber makes it difficult for ruminants to degrade and utilize [2]. Therefore, strategies are needed to improve the availability of rape straw as ruminant feed. Mixed silage of straw with materials that are rich in water soluble carbohydrates (WSC), such as whole crop corn, promises to exhibit good fermentation quality for long-term storage as a rapid decrease in pH [3,4]. The microorganism enriched during the ensiling process could break down the ligno-cellulosic bonds of the plant cell wall [5,6], which facilitates ruminal microbial colonization and thus enhances the feeding value of straw for ruminants. However, no study has yet been conducted to evaluate effects of mixed silage on the carbohydrate composition of rape straw, which is critical in improving potential utilization of rape straw.

Microorganisms colonizing the rumen play crucial roles in digesting plant materials [7], which can change with prolongation of incubation time [8]. Disparate forages can alter attached microbiota, owing to their chemical composition and ruminal availability differences [9]. For instance, rice straw treated with sodium hydroxide can enrich fibrolytic species compared to untreated rice straw [10]. The hydrolyzed plant cell walls may enhance bacterial colonization by generating substrates that favor specific microorganisms [11]. However, it remains unclear whether changes in the micro-structure of rape straw during the ensiling process could impact ruminal bacterial attachment, and consequently influence ruminal substrate degradation.

We hypothesized that mixed ensiling rape straw with whole crop corn could produce good fermentation quality, and disrupt the fiber structure of rape straw, and thus result in better nutrient and ruminal utilization value of rape straw. This study was designed to investigate fermentation quality in a mixed silage treatment, and observed the fiber structure changes of rape straw during the mixed ensiling process. An *in situ* ruminal incubation technique was employed to investigate the effects of the mixed silage treatment on ruminal degradation kinetics and the attached bacterial community.

## 2. Materials and Methods

### 2.1. Silage Preparation and Sample Collection

Whole crop corn was harvested with a stem cut about 15 cm above the ground level when 50% to 60% of the grain had the characteristics of the dough stage. Rape was harvested 30 d after the end of flowering, and rape straw was air-dried after it was obtained. Both rape straw and whole crop corn were harvested in Hulunbuir City (47°05′ N to 53°20′ N), Inner Mongolia Autonomous Region, China. Prior to ensiling, both rape straw and whole crop corn were chopped into 2–3 cm pieces. For the control treatment, a portion of whole crop corn was packed and sealed into plastic bags (20 cm × 30 cm), and anaerobically fermented for 60 d at room temperature to obtain corn silage, which was then mixed with rape straw at the dry matter (DM) ratio of 1:2 to obtain the mixture of rape straw and corn silage (control). In the preparation of mixed silage, rape straw was combined with whole crop corn at the DM ratio of 2:1. The mixture was sprayed with distilled water to adjust moisture content to around 60%, as described in our previous study [12]. Subsequently, the mixture was packed and sealed into plastic bags (20 cm × 30 cm), and anaerobically fermented for 60 d at room temperature to obtain mixed silage. Both the control and mixed silage were prepared with four replicates. Samples collected for the analysis of fermentation parameters, nutrient compositions and bacterial compositions were according to procedures outlined in our previous studies [12].

### 2.2. Animals

The experiment was conducted at the beef cattle farm of Guangxi Taiminxing Animal Husbandry development Co., Ltd., Nanning, China. Three ruminally cannulated Holstein bulls (12 mo, 280 ± 7.5 kg) were employed for the *in situ* incubation. Cattle were offered a high-forage diet consisting of wheat straw and concentrates (Table 1), and fed twice daily at 0800 and 1700 in individual stalls.

### 2.3. In Situ Ruminal Incubation

Holstein bulls were given 30 d to adapt to the diet and environment before the *in situ* ruminal incubation. The procedure of incubation with nylon bags in the rumen of animal was slightly modified according to the procedure described by Zhang et al. [14]. Briefly, approximately 6.5 g of substrate (either control or mixed silage) was weighed into polyester bags. For each Holstein bull, a total of 20 bags were prepared for each forage, with 16 bags allocated for degradation kinetics analysis and 4 bags for assessing microbial community changes. Incubation was staggered in time to ensure no more than 12 bags were present in the rumen for each Holstein bull at each time point, ensuring adequate contact between the nylon bags and rumen microbes. For the degradation kinetics analysis, incubation times were set at 4, 12, 24, 48, 72, 96, 120 and 216 h, with duplicate bags collected at each sampling time. For the microbial community analysis, incubation times were set at 4 and 48 h, with duplicate bags collected at each time point. Upon removal from the rumen, bags for degradation kinetics analyses were put into ice water to arrest microbial activity and then dried at 65 °C for the subsequent chemical analysis. Microbial bags were cleaned with phosphate buffer saline (pH = 7.4), and the liquid was gently wrung out. Then, the inner content of the bags was put into a 15-mL tube and stored at −80 °C for microbial DNA extraction.

### 2.4. Analytical Methods

The DM and crude protein (CP) were determined as outlined in AOAC methods [15]. NDF and ADF were measured as outlined in Van Soest et al. [16], and hemicellulose content was calculated as NDF minus ADF. Scanning electron microscopy (model SU8010, Hitachi, Tokyo, Japan) was employed to obtain images of rape straw according to the manufacturer’s instructions. Lactate was measured according to the method described by Taylor [17]. Individual VFA concentrations were measured using a gas chromatograph (Agilent 7890 A, Agilent Inc., Palo Alto, CA, USA) [18]. The ammonia-N concentration was measured using the phenol–hypochlorite colorimetric method [19].

### 2.5. DNA Extraction and High-Throughput Sequencing

DNA extraction was performed using methodology modified from that of Ma et al. [20]. Illumina sequencing was performed using the MiSeq platform (Illumina, San Diego, CA, USA) at the Shanghai Biozeron Biological Technology Co., Ltd. (Shanghai, China). After PCR amplification, all amplicon libraries were sequenced and the barcodes and sequencing primers were removed before data processing. Raw reads were processed using SMRT Link Analysis software version 11.0 to obtain demultiplexed circular consensus sequence (CCS) reads with the following settings: minimum number of passes = 3, minimum predicted accuracy = 0.99. Raw reads were processed through SMRT Portal to filter sequences for length (>1000 or <1800 bp) and quality. Sequences were further filtered by removing barcode and primer sequences with Lima Pipeline (Pacific Biosciences demultiplexing barcoded software, https://lima.how/). The circular consensus sequencing reads were clustered into amplicon sequences variants using QIIME 2-data2 [21]. The phylogenetic affiliation of each 16S rRNA gene sequence was analyzed by RDP Classifier (https://mothur.s3.us-east-2.amazonaws.com/wiki/trainset19_072023.rdp.tgz, accessed on 15 July 2025) against the SILVA (SSU138) 16S rRNA database using a confidence threshold of 70% [22].

### 2.6. Calculations

The exponential model of Ørskov and McDonald [23] was used to analyze degradation kinetics parameters. The effective rumen degradation (*ED*, g/kg) was calculated according to the method described by McDonald [24]. The undegradable fraction of NDF (μNDF, g/kg) was estimated by 1000 minus NDF disappearance measured at 216-h incubation time.

### 2.7. Statistical Analysis

Data was analyzed using SPSS 26.0 software (Chicago, IL, USA). Exponential parameters were tested by paired *T*-tests based on each animal. Nutrient disappearances at each incubation time were analyzed using a linear mixed model with treatment (*n* = 2) as fixed effect and animal (*n* = 3) as random effect. A Wilcoxon rank-sum test was employed for analyzing attached bacteria compositions. Principal coordinate analysis (PCoA) using amplicon sequences variants with adonis in vegan of R 4.0.3 based on the Bray–Curtis distance was used to test effects of treatment and incubation time on the attached bacterial community in the substrate. *p* ≤ 0.05 indicates that a finding was significant, and 0.05 < *p* ≤ 0.10 indicates a tendency to be significant.

## 3. Results

Mixed silage exhibited good silage quality, as indicated by the low pH (4.22), ammonia-N (2.66 g/kg), propionate (2.29 g/kg) and butyrate (0.01 g/kg), and good lactate (32.3 g/kg) (Table 2). The mixed silage was dominated by the genuses of *Lentilactobacillus* (52.6%) and *Pediococcus* (23.6%) (Table 2).

Compared with the control group, mixed silage had lower NDF (623 vs. 678 g/kg), ADF (413 vs. 440 g/kg) and hemicellulose (210 vs. 238 g/kg) contents, and higher NDS content (377 vs. 322 g/kg) (Table 2). Mixed silage altered the physical structure of rape straw, as indicated by the scanning electron microscopy images (Figure 1). The surface of rape straw was smooth and complete in the control group, while the fiber surface connection in mixed silage was broken, leading to partially regular or irregular pore structures with large numbers of microbes colonized in.

In the present study, mixed silage had distinct kinetics of DM or NDF disappearance compared to those of the control, and exhibited greater (*p* ≤ 0.05) DM or NDF disappearance at the incubation times of 4, 120 and 216 h than the control (Figure 2).

The control and mixed silage showed distinct *in situ* kinetics of ruminal disappearance in DM and NDF (Table 3). Mixed silage had greater (*p* ≤ 0.05) *a*, *a* + *b*, *ED_2_* and *ED_6_* for both *in situ* DM and NDF disappearance. Mixed silage had greater (*p* ≤ 0.05) *b* for *in situ* NDF disappearance, and tended to have greater *b* and lower lag for *in situ* DM disappearance (0.05 < *p* ≤ 0.10). Mixed silage exhibited greater (*p* ≤ 0.05) *ED* and lower (*p* ≤ 0.05) μNDF than the control group.

The bacterial communities attached to the incubated substrate were greatly affected by the incubation time, independent of the treatment. Mixed silage had a similar relative abundance of bacteria at phylum and genus levels (*p* > 0.10), when compared with the control group (Table 4). Extending incubation time from 4 to 48 h enriched the phyla Proteobacteria, Fibrobacteres and Actinobacteria, and genuses *Kineothrix*, *Lacrimispora*, *Negativibacillus*, *Stenotrophomonas* and *Fibrobacter* (*p* ≤ 0.05), and decreased the phylum Bacterodietes, and genuses *Prevotella* and *Paraprevotella* (*p* ≤ 0.05) (Table 4). The PCoA analyses showed that incubation time, rather than treatment, led to the distinct clustering, with PC 1 and PC 2 explaining 30% and 23% of variation in bacterial amplicon sequences variants, respectively (Figure 3, *P*_Incubation time_ ≤ 0.05).

## 4. Discussion

Farmers quickly evaluate the quality of silage by its appearance, color, smell, water content and pH [25,26]. It has been reported that a suitable water content of silage can reduce the likelihood of bad fermentation, and that silage ferments well when the pH is below 4.2 [4,27]. Lactic acid bacteria in silage can use carbohydrates to produce lactate, which reduces pH value and inhibits the multiplication of other microorganisms [28,29]. Butyrate is a product of poorly fermented silage and carries the risk of causing clinical ketosis in livestock, and was present at a low level as 0.01 g/kg in the product of mixed silage. The pH, desirable lactate content and low butyrate content indicated that the mixed silage we produced was of good quality.

Enrichment of homofermentative *Pediococcus* [30] and heterofermentative *Lentilactobacillus* [31,32] contributed to the favorable fermentative quality in the mixed silage. *Erwinia*, *Pantoea* and *Stenotrophomonas* are widely distributed in soil and plants [33,34,35], and their appearance in the mixed silage might be due to the production process of rape straw. In summary, mixed silage exhibited good fermentation quality by enriching *Lentilactobacillus* and *Pediococcus*.

Silage could effectively reduce the fiber content in crop straw. Yang et al. [36] have reported that the NDF and ADF content in *Leymus chinensis* is decreased after ensiling. A previous study showed that silage treatment decomposed the platelet structure of epicuticular wax crystals on the leaves of fresh alfalfa [37], so the fermentation of microorganisms in the mixed silage could impact on the micro-structure of forage. Similarly, the mixed silage treatment reduced the NDF and ADF content of crop straw in the present study, as the surface of rape straw was destroyed. This change comes from the microorganisms that decompose and ferment substances in straw to obtain energy and nutrients during the ensiling process [38].

Disruption of fiber structure helps to partially hydrolyze hemicellulose and solubilize lignin, leading to an increased surface for the colonization of ruminal microorganisms [39], and thus facilitates rumen microbial degradation, especially for complicated plant cell carbohydrates [40,41], which is important for the utilization of straw [38]. That is also the reason for the greater ruminal degradation of mixed silage in the present study. Such enhanced ruminal microbial degradation can be attributed to the destruction of rape straw during the ensiling process. In future, an in vivo study will be conducted to further analyze the effects of silage feeding on production parameters.

Mixed silage had greater *a* for both DM and NDF disappearance in our study. The greater *a* in the mixed silage could be attributed to the greater organic acids and monosaccharides contents. Similar findings were reported by Thomas et al. [42], who observed that the greater washout fractions in sorghum silage are associated with higher water-soluble carbohydrate concentrations. Furthermore, the damage to the surface of rape straw during the ensiling process may lead to the detachment of minute fragments, allowing them to pass through the aperture of the polyester bags, leading to greater rapidly soluble and washout nutrient fractions in the mixed silage. Mixed silage enhanced ruminal NDF availability in the present study, which was consistent with findings observed by Beauchemin et al. [43]. However, some studies have suggested that the reduction in structure carbohydrate content of straw may lead to decreased fiber disappearance [44]. Although straw treatments can increase the fraction of the more readily available portion of the NDF [45], they may also result in a greater proportion of undegradable fiber in NDF residue [46]. In the current study, mixed ensiling demonstrated the ability to reduce undegradable fiber in rape straw, as evidenced by the increased *a* + *b* and *ED*.

Bacteria play important roles in the carbohydrate degradation and utilization of straw by attaching to the surface [41,47]. The change of remaining biodegradable nutrients in the nylon bags due to pasture degradation will select the microorganism attached to it [9]. In our study, the mixed silage treatment scarcely altered the bacteria composition attached to the incubated substrate. Pu et al. [48] also reported that the bacterial community attached was not affected by the type of straw (rice or wheat straw). However, Liu et al. [9] reported that the microbial community attached was strongly affected by the straw type (rice straw or alfalfa hay). These different results may be partly explained by differences in the content of chemical ingredients and availability between the straws. The enhanced rumen degradation observed for mixed silage may be attributed to easier access of carbohydrate enzymes resulting from the disruption of rape straw during the ensiling process, rather than relying on the bacterial attachment. Incubation time was another factor affecting ruminal microbiota. The phylum Fibrobacteres is adept at degrading plant cell walls [49], facilitating the degradation of intricate crystalline regions of straw in later incubation stages [41,47]. Both the phyla Bacteroidetes and the genus *Prevotella* are adept at colonizing and utilizing soluble carbohydrates in the early incubation period [8]. That is the reason for the enrichment of the phyla Bacteroidetes and genus *Prevotella* in the early incubation period, and the phyla Fibrobacteres in the later incubation time in the present study. Furthermore, 48 h of incubation time also enriched the genuses *Stenotrophomonas*, *Papillibacter*, *Kineothrix*, *Lacrimispora* and *Negativibacillus*, which might also contribute to the degradation of structural carbohydrate and warrant further investigation.

## 5. Conclusions

Mixed silage disrupts the fiber structure of rape straw, resulting in higher NDS content and lower fiber content in the mixed silage. This treatment enhances *in situ* ruminal DM and NDF degradation by a creating higher total potentially degradable fractions in the mixed silage. Such enhanced rumen degradation is not associated with changes in attached bacterial communities. Instead, incubation time had a great influence on the bacterial communities, enriching the phyla Fibrobacteres and the genuses *Stenotrophomonas*, *Papillibacter*, *Kineothrix*, *Lacrimispora* and *Negativibacillus* at 48 h of incubation. In summary, the mixed silage treatment improve the feed utilization of rape straw by creating easier physical accessibility.

## Figures and Tables

**Figure 1 animals-15-02131-f001:**
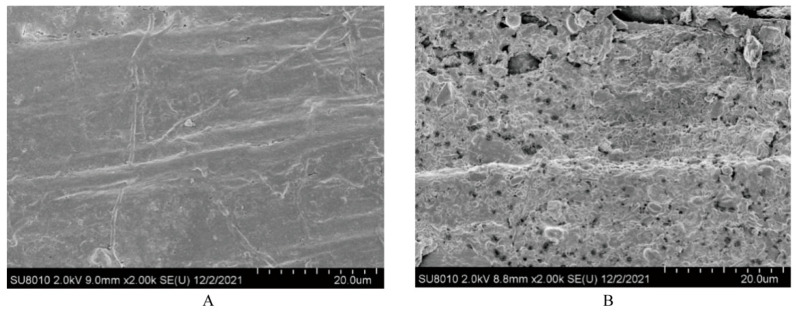
Scanning electron microscopy images (scale bar = 20 μm) of rape straw in treatments without (i.e., control, (**A**)) or with mixed ensiling (i.e., mixed silage, (**B**)). Control, a mixture of rape straw and whole crop corn silage; Mixed silage, mixed silage of rape straw and whole crop corn.

**Figure 2 animals-15-02131-f002:**
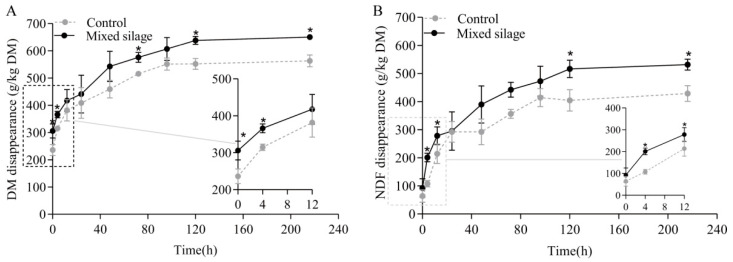
Effects of mixed silage on the kinetics of *in situ* ruminal disappearance of DM (**A**) and NDF (**B**). Control, a mixture of rape straw and whole crop corn silage; Mixed silage, mixed silage of rape straw and whole crop corn. * indicates significant difference (*p* ≤ 0.05).

**Figure 3 animals-15-02131-f003:**
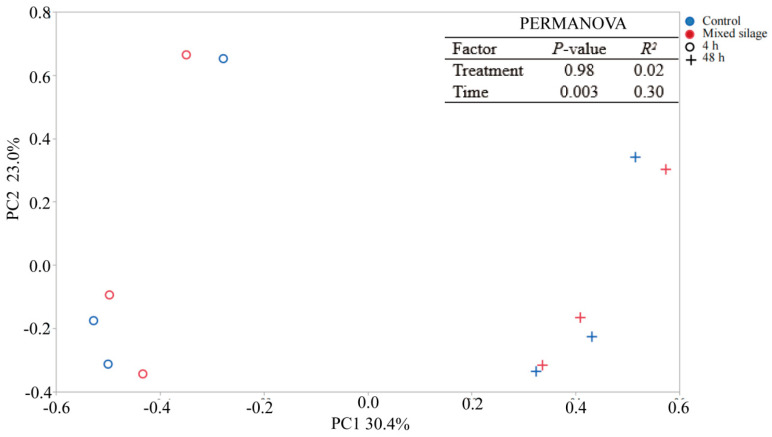
Principal coordinate analysis (PCoA) of bacterial community attached to control and mixed silage at 4-h and 48-h incubation. Control, a mixture of rape straw and whole crop corn silage; Mixed silage, mixed silage of rape straw and whole crop corn.

**Table 1 animals-15-02131-t001:** The ingredients and nutrient composition of the cattle diet (g/kg DM).

Item *	Value
Ingredients, g/kg	
Wheat straw	800
Corn	115.9
Soybean meal	13.3
Cottonseed meal	15.2
Wheat bran	15.2
Soybean hull	11.4
DDGS	9.5
NaHCO_3_	1.9
NaCl	1.9
Premix	5.7
Urea	10.0
Chemical composition, g/kg	
DM	921
OM	902
CP	82
NDF	634
ADF	377
NEm, MJ/kg	4.19
NEf, MJ/kg	2.18

* DM, dry matter; OM, organic matter; CP, crude protein; NDF, neutral detergent fiber; ADF, acid detergent fiber; NEm, net energy for maintenance; NEf, net energy for fattening. The premix provided vitamins and microelements per kilogram as follows: 1,000,000 IU of vitamin A; 200,000 IU of vitamin E; 8000 mg of Zn; 80 mg of Se; 120 mg of I; 2000 mg of Fe; 40 mg of Co; 2500 mg of Mn; 2000 mg of Cu. The NEm and NEf were calculated according to Feeding Standard of Beef Cattle (NY/T 815-2004, China [13]).

**Table 2 animals-15-02131-t002:** Fermentation parameters and bacterial compositions in the control and mixed silage.

Items *	Control	Mixed Silage
Fermentation end products, g/kg DM		
pH	-	4.22 ± 0.15
Ammonia-N, g/kg DM	3.52 ± 0.32	2.66 ± 0.18
Lactate, g/kg DM	20.5 ± 1.22	32.3 ± 1.36
Acetate, g/kg DM	9.23 ± 1.23	24.2 ± 1.56
Propionate, g/kg DM	0.26 ± 0.19	2.29 ± 0.22
Butyrate, g/kg DM	0.05 ± 0.02	0.01 ± 0.01
Nutrient composition, g/kg DM		
DM	408 ± 1.25	405 ± 2.00
CP	60.5 ± 0.78	61.3 ± 0.89
NDS	322 ± 2.12	377 ± 1.56
NDF	678 ± 2.12	623 ± 1.56
ADF	440 ± 1.56	413 ± 0.98
Hemicellulose	238 ± 1.76	210 ± 1.35
Bacterial compositions (genus), %		
*Lentilactobacillus*	-	52.6 ± 2.35
*Pediococcus*	-	23.6 ± 2.89
*Pantoea*	-	5.29 ± 1.23
*Stenotrophomonas*	-	3.23 ± 0.98
*Levilactobacillus*	-	2.34 ± 0.78
*Erwinia*	-	2.17 ± 1.35
*Secundilactobacillus*	-	1.23 ± 0.46
*Escherichia*	-	0.82 ± 0.50
*Salmonella*	-	0.61 ± 0.23
*Sphingobacterium*	-	0.61 ± 0.12
*Sarcina*	-	0.53 ± 0.13
*others*	-	6.97 ± 2.67

* DM, dry matter; CP, crude protein; NDS, neutral detergent solubles; NDF, neutral detergent fiber; ADF, acid detergent fiber. Control, a mixture of rape straw and whole crop corn silage; Mixed silage, ensiling treatment of rape straw and whole crop corn mixture; -, not measured. The same as below.

**Table 3 animals-15-02131-t003:** Effects of mixed ensiling on the kinetics of *in situ* ruminal disappearance of DM and NDF.

Items ^a^	Treatments ^b^		SEM	*p*-Value
	Control	Mixed Silage		
DM				
*a*, g/kg	289	340	12.6	0.01
*b*, g/kg	272	318	13.0	0.08
*a* + *b*, g/kg	561	659	22.3	0.01
*c*,% h^−1^	3.18	2.37	0.49	0.29
*Lag*, h	2.82	2.46	0.232	0.08
*ED_2_*, g/kg	449	507	17.5	0.02
*ED_6_*, g/kg	379	427	16.2	0.03
NDF				
*a*, g/kg	108	159	12.1	0.01
*b*, g/kg	310	370	14.5	0.04
*a* + *b*, g/kg	418	529	26.0	0.01
*c*,% h^−1^	3.16	2.51	0.42	0.31
*lag*, h	3.13	2.36	0.292	0.31
*ED_2_*, g/kg	293	361	19.4	0.02
*ED_6_*, g/kg	212	266	16.4	0.04
μNDF, g/kg	582	471	26.0	0.01

^a^ *a* soluble and washout nutrient fractions and represents the rapidly degraded component; *b*, potentially degradable nutrient fractions and represents the slowly degraded component; *a* + *b*, total potentially degradable fraction; *c*, disappearance rate of fraction *b*; *lag*, lag time before the initial degradation of fraction *b*; *ED*_2_ and *ED*_6_, effective ruminal degradation calculated with passage rate being 0.02/h and 0.06/h respectively; µNDF, undegraded fraction of NDF at 216 h of incubation time. ^b^ Control, a mixture of rape straw and whole crop corn silage; Mixed silage, mixed ensiling treatment of rape straw and whole crop corn mixture.

**Table 4 animals-15-02131-t004:** Bacterial compositions with relative abundance larger than 0.5% attached to incubated substrate in control or mixed silage.

Items *	Treatment		Incubation Time (h)		SEM	*p*-Value		
	Control	Mixed Silage	4	48		Treatment	Incubation Time	Treatment × Incubation Time
Phylum Firmicutes	43.0	45.5	43.5	45.0	1.12	0.33	0.55	0.48
*Papillibacter*	2.29	2.31	0.98	3.62	0.414	0.92	<0.001	0.47
*Kineothrix*	1.46	1.69	1.29	1.86	0.151	0.29	0.03	0.06
*Lacrimispora*	1.58	2.11	1.06	2.63	0.351	0.26	0.01	0.27
*Negativibacillus*	0.59	0.54	0.29	0.83	0.120	0.70	0.01	0.66
Phylum Bacteroidetes	32.2	31.2	38.7	24.6	2.27	0.59	<0.001	0.66
*Prevotella*	16.3	16.8	23.0	10.2	2.36	0.88	0.01	0.93
*Paraprevotella*	5.89	4.73	7.12	3.50	0.752	0.21	0.01	0.55
Phylum Proteobacteria	16.4	14.9	11.3	20.0	1.50	0.23	<0.001	0.73
*Stenotrophomonas*	9.98	8.54	4.54	14.0	1.51	0.14	<0.001	0.13
Phylum Fibrobacteres	1.79	2.00	0.73	3.06	0.459	0.63	0.001	0.15
*Fibrobacter*	1.75	1.94	0.68	3.01	0.458	0.66	0.001	0.14
Phylum Actinobacteria	0.61	0.49	0.30	0.79	0.094	0.37	0.01	0.93

* Control, a mixture of rape straw and whole crop corn silage; Mixed silage, mixed silage of rape straw and whole crop corn.

## Data Availability

The data presented in this study are available upon request from the corresponding author.

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
