# Peer review of "Mixed Ensiling Increases Degradation Without Altering Attached Microbiota Through In Situ Ruminal Incubation Technique"

_animals, 2025, doi:10.3390/ani15142131_

Round 1

Reviewer 1 Report

Comments and Suggestions for Authors

Dear authors, Line 33 to 35: standardize is silage or ensilage? Let's see that there are differences between these 2 terms. Line 41: "and tended to have lower (P ≤ 0.10) lag time for DM disappearance." It's better to remove it, a tendend is not a statement. In the abstract, the values ​​of each nutrient? It just describes that it has increased or decreased. Present and values ​​each variable.

Line 56: mixed silage 

Line 70: during ensiling or in silage?

Line 73:  "ensiling" is the process and "silage" is the feed. Rewie all text.

Line 97: The approval of the use of animals for the experiment?

Line 237: "the Silage"

Use more of the conclusion in the summary and abstract

Author Response

Comments 1: Dear authors, Line 33 to 35: standardize is silage or ensilage? Let's see that there are differences between these 2 terms. Line 41: "and tended to have lower (P ≤ 0.10) lag time for DM disappearance." It's better to remove it, a tended is not a statement. In the abstract, the values of each nutrient? It just describes that it has increased or decreased. Present and values each variable.

Response 1: The standardize is silage, as treatments was mixed silage that composed of rape straw and whole crop corn with ensiling for 30 days. “and tended to have lower (P ≤ 0.10) lag time for DM disappearance.” the sentence was removed in lines 45. The changed values of each nutrient was supplemented as “Mixed ensiling disrupted the fiber structure of rape straw, thus had lower fiber contents compared to control, as NDF and ADF contents ‌decreased by 55 g/kg‌ (678 vs 623 g/kg) ‌and 27 g/kg‌ (440 vs 413 g/kg), respectively. Compared to control group, ruminal DM disappearance of mixed silage significantly (P ≤ 0.05) increased from 315 to 366 g/kg (+16.2%) at incubation time of 4 h, 552 to 638 g/kg (+15.6%) at 120 h, and 563 to 651 g/kg (+15.6%) at 216 h. Similarly, compared to control group, NDF disappearance of mixed silage significantly (P ≤ 0.05) rose from 112 to 201 g/kg (+79.5%) at 4 h, 405 to 517 g/kg (+27.7%) at 120 h, and 429 to 532 g/kg (+24.0%) at 216 h. Compared to control group, soluble and washout nutrient fractions (a) of DM or NDF fraction in mixed silage significantly (P ≤ 0.05) rose from 289 to 340 g/kg (+17.6%), potentially degradable fractions (b) of NDF increased from 310 to 370 g/kg (+19.4%), and undegraded fraction of NDF (μNDF) decreased from 582 to 471 g/kg (-19.1%). ”in lines 41-52.

Comments 2: Line 56: mixed silage

Response 2: corrected as mixed silage in line 64.

Comments 3: Line 70: during ensiling or in silage?

Response 3: it was during ensiling process. This sentence implied that the micro-structure of rape straw could be disrupted during ensiling process.

Comments 4: Line 73:  "ensiling" is the process and "silage" is the feed. Review all text.

Response 4: revised through all the manuscript.

Comments 5: Line 97: The approval of the use of animals for the experiment?

Response 5: The approval of the use of animals for the experiment was stated in lines 312-314.

Comments 6: Line 237: "the Silage"

Response 6: corrected as mixed silage in line 291.

Comments 7: Use more of the conclusion in the summary and abstract.

Response 7: More of the conclusion in the summary was added in lines 297-299.

Reviewer 2 Report

Comments and Suggestions for Authors

The aim of the study is to evaluate the effect of mixed silage of rapeseed and corn straw on rumen utilization, in particular on fiber degradation and bacterial colonization during in situ rumen incubation.
In the abstract clarify the issue of control and sample. The control refers to a mixture already ensiled, while the sample refers to a mixture that will be subjected to ensiling.

Material and methods
2.3 The bibliographic reference for in situ incubation tests is missing.

For the microbiological part, it would be useful to indicate:
- the total number of reads per sample;
- the library coverage;
- possible removal of chimeras or singletons

2.7 it is not clear whether the animal effect was considered in the statistical processing

Table 2 indicate that the mean value and standard deviation are reported

tables 3 and 4 - improve the legend (for example SEM not reported)

Results
The microbiological results are interesting but not very detailed: the functional role of the observed bacterial genera could be better discussed.
It is said that mixed ensiling does not change the adherent bacterial community, but this is a negative data compared to the initial hypothesis, it is necessary to discuss it better: "the greater degradation is not attributable to microbial colonization but to physical accessibility".

The statistically tending differences (P ≤ 0.10) should be treated with caution as they are not significant.

There is no explicit comparison with other works that have used similar pretreatments (NaOH, enzymes, ammonia).

Expanding the discussion, it is a bit poor

The conclusions are a bit too synthetic, for example the implication of the results obtained is missing

Author Response

Comments 1: The aim of the study is to evaluate the effect of mixed silage of rapeseed and corn straw on rumen utilization, in particular on fiber degradation and bacterial colonization during in situ rumen incubation.

In the abstract clarify the issue of control and sample. The control refers to a mixture already ensiled, while the sample refers to a mixture that will be subjected to ensiling.

Response 1: the treatments was more clearly clarified as “The experiment comprised two treatments: the mixture of rape straw and corn silage (control), and mixed silage treatment of rape straw and whole-crop corn (mixed silage)” in lines 36-37.

Material and methods

Comments 2: 2.3 The bibliographic reference for in situ incubation tests is missing.

Response 2: reference was added in line 117, 351-353.

Comments 3: For the microbiological part, it would be useful to indicate:

- the total number of reads per sample;

- the library coverage;

- possible removal of chimeras or singletons

Response 3: information was added in lines 153-159.

Comments 4: 2.7 it is not clear whether the animal effect was considered in the statistical processing

Response 4: the animal was considered as random effect in the statistical processing, which was more clearly clarified in lines 173-174.

Comments 5: Table 2 indicate that the mean value and standard deviation are reported, tables 3 and 4 - improve the legend (for example SEM not reported)

Response 5: for table 2, the values of fermentation endproducts, nutrient composition and bacterial composition were reported as the mean value and standard deviation, which was not statistically analyzed. While, for table 3 and 4, statistically analysis were employed, therefore, the results used SEM.

Results

Comments 6: The microbiological results are interesting but not very detailed: the functional role of the observed bacterial genera could be better discussed.

Response 6: The microbiological results are expanded in the section of Results and Discussion in lines 192-198.

Comments 7: It is said that mixed ensiling does not change the adherent bacterial community, but this is a negative data compared to the initial hypothesis, it is necessary to discuss it better: "the greater degradation is not attributable to microbial colonization but to physical accessibility".

Response 7: physical accessibility that accounts for the greater degradation in the mixed silage was emphasized in lines 217-226 and 266-269.

Comments 8: The statistically tending differences (P ≤ 0.10) should be treated with caution as they are not significant.

Response 8: The statistically tending differences were removed through the whole manuscript.

Comments 9: There is no explicit comparison with other works that have used similar pretreatments (NaOH, enzymes, ammonia).

Response 9: the manuscript with similar pretreatments was referred and discussed in lines 180-209.

Comments 10: Expanding the discussion, it is a bit poor

Response 10: the discussion was expanded in lines180-282.

Comments 11: The conclusions are a bit too synthetic, for example the implication of the results obtained is missing

Response 11: Supplemented in lines 297-299.

Reviewer 3 Report

Comments and Suggestions for Authors

The problem of utilization of rapeseed straw and its low nutritional value is an important issue, which justifies the purpose of the study.

The proposal to mix the straw with crops that are commonly grown and rich in carbohydrates like corn is in line with current trends in research to improve forage quality.

The introduction and hypotheses and the objective are well written

Material and methods
No imformation about the ethics committee. With conjugated animals there must be approval

At what latitudes were canola and corn harvested?

Was any additive used in the ensiling process ?

How were the meat cattle raised ? I don't understand Holstein meat cattle ???? Was it a mixture of meat cattle and Holstein cattle ???? Typical Holstein cattle are dairy.

Please wrap up the VFA and ammonia analysis carefully. The literature provided is not sufficient because it mainly describes methane emissions
The summary is good

Please add that the study should be conducted in vivo and analyze the effect of silage feeding on production parameters

Results mixed up with discussion
No separate discussion chapter
poorly written chapter
Arrange arguments around impact of incubation time vs. impact of ensiling.

Discussion conducted on 10 papers? In fact, there is no discussion. As for me, only the results are described.

Only after writing a separate discussion can the manuscript be evaluated

Author Response

The problem of utilization of rapeseed straw and its low nutritional value is an important issue, which justifies the purpose of the study.

The proposal to mix the straw with crops that are commonly grown and rich in carbohydrates like corn is in line with current trends in research to improve forage quality.

The introduction and hypotheses and the objective are well written

 Material and methods

Comment 1: No information about the ethics committee. With conjugated animals there must be approval

Response 1: the ethics committee was presented in the manuscript in lines 312-314.

Comment 2: At what latitudes were canola and corn harvested?

Response 2: Both rape straw and whole crop corn were harvested in Hulunbuir City (47°05′N to 53°20′N), Inner Mongolia Autonomous Region, China. The information was added in the manuscript in lines 93-95.

Comment 3: Was any additive used in the ensiling process ?

Response 3: No additives was used in the ensiling process.

Comment 4: How were the meat cattle raised ? I don't understand Holstein meat cattle ???? Was it a mixture of meat cattle and Holstein cattle ???? Typical Holstein cattle are dairy.

Response 4: Cattles were fed in individual stalls, and the information was added in line 113. Holstein bulls raised at the beef cattle farm of Guangxi Taiminxing animal husbandry development Co., Ltd., Nanning city were used in this trial. Holstein beef cattles were corrected as Holstein bulls in the manuscript.

Comment 5: Please wrap up the VFA and ammonia analysis carefully. The literature provided is not sufficient because it mainly describes methane emissions

The summary is good

Response 5: The literature provided for individual VFA analysis was corrected in line 359-361. The methods for VFA analysis were detailed as “After re-centrifuging at 15,000 g, the VFA in the supernatants were measured using gas chromatography (Agilent 7890A, Agilent Inc., Palo Alto, CA). The acids were separated with a DB-FFAP column (30 m × 0.25 mm i.d., 0.25 μm mesh), and detected with an R flame ionisation detector (FID).The carrier gas was nitrogen at a rate of 0.8 ml/min. The analysis was initially isothermal for 2 min at 60 °C and then increased to 220 â„ƒ at a rate of 20 â„ƒ min-1, with a detector temperature of 280 â„ƒ. VFA were identified and quantified from chromatograph peak areas using calibration with external standards.”

Comment 6: Please add that the study should be conducted in vivo and analyze the effect of silage feeding on production parameters

Response 6: Added in lines 225-226, as “Furthermore, in vivo study would be conducted to further analyze the effect of silage feeding on production parameters.”.

Comment 7: Results mixed up with discussion

No separate discussion chapter

poorly written chapter

Arrange arguments around impact of incubation time vs. impact of ensiling.

Response 7: Results were combined with discussion, and corrected in lines 180-282.

Comment 8: Discussion conducted on 10 papers? In fact, there is no discussion. As for me, only the results are described.

Only after writing a separate discussion can the manuscript be evaluated

Response 8: I am so sorry for the missed discussion. In fact, results and discussion was combined in our manuscript, but the discussion was a bit poor. I have corrected the section of “results and discussion” in lines 180-282.

Round 2

Reviewer 3 Report

Comments and Suggestions for Authors

The manuscript is suitable for publication after splitting Chapter 3
into Chapter 3 Results
Chapter 4 Discussion

Author Response

Comments 1: The manuscript is suitable for publication after splitting Chapter 3
into Chapter 3 Results
Chapter 4 Discussion

Response 1: Dear reviewers, Chapter 3 was split into Chapter 3 Results and Chapter 4 Discussion.